# Function-as-a-Service: From an Application Developer's Perspective

Ali Raza
*Boston University*
*araza@bu.edu*

Ibrahim Matta
*Boston University*
*matta@bu.edu*

Nabeel Akhtar
*Akamai Technologies Inc.*
*nakhtar@akamai.com*

Vasiliki Kalavri
*Boston University*
*vkalavri@bu.edu*

Vatche Isahagian
*IBM Research*
*vatchei@ibm.com*

## Abstract

In the past few years, FaaS has gained significant popularity and became a go-to choice for deploying cloud applications and micro-services. FaaS with its unique 'pay as you go' pricing model and key performance benefits over other cloud services, offers an easy and intuitive programming model to build cloud applications. In this model, a developer focuses on writing the code of the application while infrastructure management is left to the cloud provider who is responsible for the underlying resources, security, isolation, and scaling of the application. Recently, a number of commercial and open-source FaaS platforms have emerged, offering a wide range of features to application developers. In this paper, first, we present measurement studies demystifying various features and performance of commercial and open-source FaaS platforms that can help developers with deploying and configuring their serverless applications. Second, we discuss the distinct performance and cost benefits of FaaS and interesting use cases that leverage the performance, cost, or both aspects of FaaS. Lastly, we discuss challenges a developer may face while developing or deploying a serverless application. We also discuss state of the art solutions and open problems.

## 1 Introduction

Function-as-a-Service (FaaS) has emerged as a new paradigm that makes the cloud-based application development model simple and hassle-free. In the FaaS model, an application developer focuses on writing code and producing new features without worrying about infrastructure management, which is left to the cloud provider.[1] FaaS was first introduced by Amazon in 2014 as AWS Lambda [3], and since then, other commercial cloud providers have introduced their serverless platforms, i.e. Google Cloud Function (GCF) [22] from Google, Azure Function [14] from Microsoft, and IBM Cloud Function [23] from IBM. There are also several open-source projects like Apache OpenWhisk, Knative, OpenLambda, Fission, and others.

At the time of the inception of the Internet, applications were built and deployed using dedicated hardware acting as servers, which needed a high degree of maintenance and often lead to under-utilization of resources [60, 61]. Moreover,

---

[1]We assume FaaS is a computing service provided by a serverless platform managed by a cloud provider.

adding/removing physical resources to scale to varying demand, and debugging an application, was a cumbersome task. Under-utilization of resources and higher cost of maintenance led to the invention of new technologies like virtualization and container-based approaches. These approaches not only increased resource utilization but also made it easy to develop, deploy, and manage applications. Many tools [60, 61, 79, 119] were built to help users orchestrate resources and manage the application. Although virtualization and container-based approaches lead to higher utilization of resources and ease of building applications, developers still have to manage and scale the underlying infrastructure of an application, i.e. virtual machines (VMs) or containers, despite the availability of a number of approaches that would perform reactive or predictive scaling [47, 69, 92, 98, 105, 128]. To abstract away the complexities of infrastructure management and application scaling, serverless computing emerged as a new paradigm to build, deploy, and manage cloud applications. The serverless computing model allows a developer to focus on writing code in a high-level language (as shown in Table 1) and producing new features of the application, while leaving various logistical aspects like the server configuration, management, and maintenance to the FaaS platform [122].

Even though FaaS has been around for only a few years, this field has produced a significant volume of research. This research addresses various aspects of FaaS from benchmarking/improving the performance of various FaaS platforms/applications, porting new applications into a serverless model, to suggesting altogether new serverless platforms. As serverless computing is still an evolving field, there is a significant need for systematization of the knowledge (SoK) particularly from the perspective of an application developer. We believe that for an application developer, an ideal SoK paper should address three main aspects: 1) current state of FaaS platforms, *e.g.* performance and features, 2) what makes serverless computing ideal for certain classes of applications, and 3) and future research directions for helping a developer leverage the full potential of FaaS with her limited control over the FaaS platform.

Previous SoK papers are generally written from the perspective of the service provider. Castro et al. [56] present an overview of serverless computing and discuss the serverless architecture, development, and deployment model. Hellerstein et al. [77], Jonas et al. [82] and Baldini et al. [52] also provide an overview of serverless computing, and dis-

cuss potential challenges that a serverless provider should address for the popularization of serverless computing. Similarly, in [110], challenges and potential research directions for serverless computing are discussed. Eismann et al. [65] perform a systematic review of serverless applications and provide useful insights into the current usage of serverless platforms. Eyk et al. [118] give perspectives on how serverless computing can evolve and identify adoption challenges. Lynn et al. [90] give an overview of various features provided by popular serverless platforms. The aforementioned works generally take the perspective of a service provider and discuss the challenges and optimizations that it should introduce to improve and popularize the FaaS platform and have limited to no discussion from an application developer's perspective.

In this paper, we take a closer look at the three aforementioned aspects of FaaS from an application developer's perspective. We assess previous work related to measurements, performance improvement, and porting of applications into the FaaS computing model, and augment this with our own experimental results and insights. While we mainly take an application developer's perspective viewing the FaaS platform as a closed-loop feedback control system, we also discuss improvements and optimizations that a provider can introduce (as a developer can be a provider too in the case of open-source FaaS platforms) to provide a more holistic view to the reader. In this paper, we make the following contributions to the SoK:

- We categorize the decisions that an application developer can make during one life cycle of an application into two categories: *one-time decisions* and *online decisions*, and discuss their performance and cost implications.

- We show that the quick provisioning time, on-demand scaling, and true "pay as you go" pricing model are key factors for FaaS adoption for various classes of applications and discuss potential challenges.

- In Section 7, we discuss the challenges and open issues that a developer may face while employing FaaS for her cloud applications. We discuss building tools and strategies for FaaSification and decomposition of legacy applications to better suit the FaaS computing model, optimizing code, tuning resources for serverless applications, and usage of FaaS in conjunction with other cloud services for cost savings.

The rest of the paper is organized as follows. We first describe the FaaS computing model and its important features (Section 2). We then present a developer's view of a FaaS platform as a closed-loop feedback control system (Section 3). Next, we look at various measurement studies that investigate different aspects of commercial and open-source FaaS platforms (Section 4). Then we present an economic model of

FaaS (Section 5) and compare it with traditional Infrastructure-as-a-Service (IaaS), and identify suitable classes of applications that can leverage serverless computing for its performance/cost (Section 6). Lastly, we discuss future challenges and research directions to make FaaS adoption efficient and easy (Section 7).

## 2 Background

Serverless computing was initially introduced to handle less frequent and background tasks, such as triggering an action when an infrequent update happens to a database. However, the ease of development, deployment, and management of an application and the evolution of commercial and open-source FaaS platforms have intrigued the research community to study the feasibility of the serverless computing model for a variety of applications [72, 94, 128, 129]. Moreover, there are systems whose aim is to help developers port their applications to a serverless programming model [62, 114].

In a serverless computing model, a developer implements the application logic in the form of stateless functions (henceforth referred to as serverless functions) in the higher-level language. We show various runtimes supported by popular FaaS platforms in Table 1. The code is then packaged together with its dependencies and submitted to the serverless platform. A developer can associate different triggers with each function, so that a trigger would cause the execution of the function in a sandbox environment (mostly containers) with specified resources, i.e. memory, CPU-power, etc. The setup time of the sandbox environment is referred to as *cold start*. In a typical case, the output of the serverless function is then returned as the response to the trigger. As serverless functions are stateless, a developer has to rely on external storage (like S3 from AWS), messages (HTTP requests) or platform API [32] to persist any data or share state across function instances[2]. The serverless computing model is different from traditional dedicated servers or VMs in a way that these functions are launched only when the trigger is activated, while in the traditional model, the application is always running (hence the term "serverless").

Serverless computing abstracts away the complexities of server management in two ways. First, a developer, only writes the logic of an application in a high-level language, without worrying about the underlying resources or having to configure servers. Second, in case the demand for an application increases, a serverless platform scales up the instances of the application without any additional configuration or cost and has the ability to scale back to zero (discussed in Section 4.4). While FaaS platforms provide typical CPU and memory power to serverless applications, it is their ability to scale quickly (in orders of milliseconds) that gives them

---

[2]Note that most approaches [45, 91, 113, 116] to improve the inter-function/storage communication can only be implemented by the cloud provider.

a performance advantage over other cloud services. On the contrary, in IaaS, an application developer not only has to specify the additional scaling policies but there can be an additional cost for deploying such autoscaling services and it can take up to minutes to scale up.

In Table 1, we show some of the key features provided by popular commercial FaaS platforms[3]. While providing similar services, specific features can vary significantly from one platform to another. Generally, these platforms only allow memory as a configurable resource for the sandbox environment with the exception of GCF which also allows a developer to specify the CPU power. AWS Lambda allocates CPU in proportion to the memory allocated [6]. IBM Cloud Function seems to have a constant allocation of the CPU share regardless of the memory allocation as increasing memory does not improve runtime significantly [93]. Azure Function does not allow any configurable resource and charges the user based on the execution time and memory consumption [11]. While these platforms initially supported applications written in specific languages, they currently support more languages and custom runtimes, making it possible to run any application using FaaS.

An important feature of the serverless computing model is that serverless platforms follow the "pay as you go" pricing model. This means a user will only pay for the time a serverless function is running. This model charges a user for the execution time of the serverless function based on the resources configured for the function. A user will not be charged for deploying the function or for idle times. Even though all of the cloud providers follow a similar pricing model, the price for the unit time (*Billing Interval*) of execution can vary significantly from one cloud provider to another.

In the serverless computing model, the abstraction of infrastructure management comes at the cost of little to no control over the execution environment (and underlying infrastructure) of the serverless functions. Depending on the platform, a user can control limited configurable parameters, such as memory size, CPU power, and location to get the desired performance. Since the introduction of serverless platforms, there has been a large body of research work that aims to demystify the underlying infrastructure, resource provisioning, and eviction policies for commercial serverless platforms. Besides, these works have also looked at different aspects of performance, namely cold-starts, concurrency, elasticity, network, and I/O bandwidth shares. These research studies are helpful for the research and developer community to find a suitable serverless platform for their application and also inspire future research.

---

[3]Features listed on official documentation as of 7/30/2021.
AWS Lambda: https://aws.amazon.com/lambda
Azure Functions: https://azure.microsoft.com/services/functions
Google Cloud Functions: https://cloud.google.com/functions
IBM Cloud Functions: https://www.ibm.com/cloud/functions

## 3   Developer's View of FaaS

Serverless platforms are largely black-boxes for application developers, who submit the code of their application (with a few configurations) and in turn, the code gets executed upon the specified triggers. A user has little to no control over the execution environment, underlying resource provisioning policies, hardware, and isolation. A user has control over limited configurations through which they can control the performance of their serverless application. In what follows we categorize the decisions a developer can make for their serverless applications to get the desired performance or optimize their cost.

*One-Time Decisions:* These are the decisions that a developer can make before developing and deploying an application and include selecting the serverless platform, programming language, and location of deployment. These decisions can be dictated by the features that a serverless platform offers such as underlying infrastructure, pricing model, elasticity, or performance metrics – for example, certain languages may have lower cold-start latency or the location of deployment can affect the latency to access the application. We believe changing any of these aspects would incur significant development and deployment cost, hence a developer can make such a decision only once in the life cycle of the application.

*Online Decisions:* A developer has more freedom to change other parameters without a serious effort, including resources (memory, CPU) and concurrency limit. As we show later in this section, these parameters can affect the performance and cost of a serverless application. A developer can employ a more proactive technique to configure her serverless function based on the desired performance metric. Configuring these parameters is also important as serverless platforms provide no Service-Level Objective (SLO), i.e. guarantee on the performance of the serverless function, and a developer's only recourse to get the desired performance is through the careful configuration of these parameters. Later in Section 7, we discuss the challenges of designing proactive approaches by employing feedback control systems. These systems would continually monitor the performance of a serverless application and make these online decisions for the application, as shown in Figure 1.

There have been several measurement studies conducted by academic researchers and independent developers that have attempted to demystify different aspects of commercial and open-source serverless platforms. These studies help a developer make *one time decisions* by identifying the underlying resources, i.e. operating system, CPUs, virtualization technique, and by benchmarking various performance aspects of serverless platforms. Moreover, these studies also look at the effect of configurable parameters (*online decisions*) on the performance and cost of serverless functions establishing the need to configure these parameters carefully.

|  | AWS Lambda | Google Cloud Function | IBM Cloud Function | Microsoft Azure Function |
|---|---|---|---|---|
| Memory (MB) | {128 ... 10240} | $128 \times i$ 
 $i \in \{1,2,4,8,16,32\}$ | {128 ... 2048} | upto 1536 |
| Runtimes Supported | Node.js 14/12/10, Go 1.x, Ruby 2.7/2.5, Python 3.8/3.7/3.6/2.7, Java 11/8, .NET Core 3.1/2.1, and Custom Runtimes | Go 1.13, Python 3.9/3.8/3.7, Ruby 2.7/2.6, Java 11, Node.js 14/12/10, .NET Core 3.1, and PHP 7.4 | Node.js 12, Python 3.7/3.6, Java 8, Swift 4.2, PHP 3.7, Ruby 2.5, Go 1.15, .NET Core 2.2, and Docker | .NET Core 3.1/2.1, .NET Framework 4.8, Node.js 14/12/10/8/6, Java 11/8, PowerShell 7/6, and Python 3.9/3.8/3.7/3.6 |
| Billing | Execution time based on memory | Execution time based on memory & CPU-power | Execution time based on memory | Execution time based on memory used |
| Billing Interval | 1ms | 100ms | 100ms | 1ms |
| Configurable Resource | memory | memory & CPU-power | memory | n/a |

Table 1: Popular commercial FaaS platforms

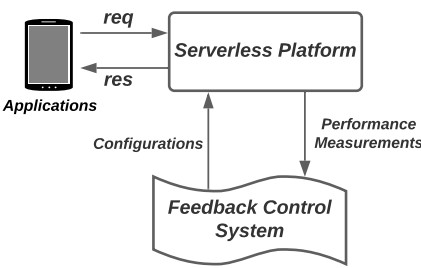

Figure 1: Feedback control systems to configure serverless functions

# 4  Measurement Studies

In Table 2, we present a classification of previous measurement studies. In this classification, we correlate the decisions (both one time and online) that a developer or a researcher can make in terms of picking the serverless platform, scripting language, and configurations, with different performance aspects, such as cold-start delay, runtime, cost, etc. Every cell in the table indicates the peer-reviewed studies that have looked at the relationship between the controlled variable (decision) and the dependent parameters (performance). In what follows, we describe in greater detail the findings of these measurement studies[4] and explain the effect of choices on different performance aspects. As our main focus in this paper is the main observations and insights of these measurement studies, we do not discuss other aspects such as the reproducibility of these studies. To this end, Scheuner and Leitner [106] present a comprehensive analysis of various measurement studies and discuss the flaws and the gaps in the FaaS measurement and benchmarking literature.

---

[4]Results of these measurement studies are subjected to changes if cloud provider decides to update hardware or other policies.

## 4.1  Cold Starts

This is perhaps the most studied aspect of serverless platforms. There have been several peer-reviewed studies attempting to quantify and remedy the effect of cold starts. The cold start comes from the fact that if a function is not being invoked recently for an amount of time set by the platform (called *instance recycling time*, and discussed later in this section), the platform destroys the sandbox environment (i.e. container) to free up the resources. On a subsequent new request, the platform will (re-)initialize the sandbox environment and execute the function, hence an extra delay would be incurred. Studies have found that cold starts can be affected by various online and one-time decisions.

- *Choice of language:* These studies show that usually, interpreted languages (Python, Ruby, Javascript) have significantly less (100x) cold-start delays as compared to compiled runtimes (Java, .NET, etc.) [17, 95, 122]. This can be due to the fact that a compiled runtime, as in Java, requires the initiation of a compute-intensive JVM, which can incur significant delay [123]. Another aspect to consider is that although interpreted languages may have less of an initial cold start, they suffer from lower execution performance compared to compiled runtimes [33].

- *Serverless provider:* Studies have shown that different providers can have different cold-start delays depending on their underlying infrastructure or resource provisioning strategy [87,95,97,122]. For example, AWS Lambda tends to place function instances from a user on the same underlying VM [97, 122], hence causing contention and increasing the cold start. Similarly, Azure instances are always assigned 1.5GB of memory, possibly increasing the cold-start time.

- *Resources:* Cold start is also impacted by the resources

| Parameters
Control $\longrightarrow$
Measured $\downarrow$ | Serverless Platform | Language | Memory/CPU | Location |
|---|---|---|---|---|
| Cold Start | [45, 87, 93, 95, 97, 122] | [93, 95, 122] | [87, 95, 122] | x |
| Cost/Performance | [45, 51, 85, 87, 93, 95, 97, 122] | [122] | [44, 88, 122, 127] | [44, 68] |
| Concurrency | [85, 87, 88, 122] | [93] | x | x |
| I/O throughput | [85, 122, 123] | x | [44, 122, 123] | x |
| Network throughput | [85, 122, 123] | x | [122] | x |
| Instance Lifetime | [88, 120, 122] | x | [122] | x |
| Underlying Infrastructure | [87, 122] | x | [88] | x |

Table 2: Measurement Studies – each cell identifies the studies establishing relation between the respective column (decision) and row (performance/platform characteristics) – 'x' means no documented relation between decision and performance

available to the function, i.e. memory/CPU [17, 95, 122]. This can be because of the fact that more resources lead to a faster setup of the execution environment [122].

- *Code Package:* Studies [17, 123] have shown that code package size, *i.e.* code and the libraries it uses, can affect the cold-start latency. This is due to the fact that the bigger the code package size, the longer it will take to load into memory [78].

The above insights can help a user develop an application in a particular language, and also configure resources based on the application's needs. If an application is latency-sensitive, a developer may choose to use a scripting language and configure more resources for the serverless function. One has to be careful with configuring more resources for the serverless function to remedy cold start, as it can increase the cost of running the serverless function as explained later in Section 4.3. Based on the finding reported in [122] on commercial serverless platforms, AWS Lambda has the least cold-start delays. Approaches to circumvent the cold start can be divided into two categories:

1) For serverless platforms: Serverless platforms can improve the cold-start latency by having fast sandboxing techniques, efficient function instance[5] placement/scheduling and by keeping the sandbox instances warm for a longer time. While the last approach can be significantly expensive for the platform as it can potentially lead to resource under-utilization (discussed in more detail in Section 4.2), there has been a significant body of research focused on improving the cold-start latency through the first two. Advanced container-management/sandboxing techniques [45, 54, 63, 97, 100, 113] employ container reuse, loose isolation between function instances, and memory snapshotting and restoring to achieve a cold-start latency that is as low as 10ms or less [113]. Other approaches suggest optimized routing schemes [43], package-aware scheduling [49], efficient capacity planning [74] and reuse of resources [115] to reduce the cold-start latencies.

---
[5]Function instance refers to the sandbox environment executing the code of a serverless function.

2) For the developers: The aforementioned fast sandboxing approaches will only work if a developer has complete control over the serverless platform. In case a developer is using a commercial serverless platform, their approach to mitigate cold start will be different. In addition to carefully selecting the language and serverless platform to develop and deploy their application based on previous findings, they can also control cold start through carefully configuring resources for the application. There are several articles published [2, 18, 30, 40, 86], which suggest certain design changes in the application to avoid unnecessary cold starts such as sending dummy requests to the serverless function that perform early *exit* without performing any computation. While these approaches may keep the function warm, they can also introduce extra cost (discussed in Section 4.3) as there is a fixed cost charged for each request and some FaaS platforms round up the execution time to the nearest 100ms, so even if the function performs early *exit*, the user would be charged some cost. A recent feature from FaaS platforms, such as AWS Lambda [36] and Azure Function [10], allows their user to specify a minimum number of function instances to be kept warm all the time to avoid unnecessary cold starts but a user is charged for enabling this feature.

> *Summary:* Cold start can be impacted by the virtualization techniques and function eviction policies employed by the serverless platform. From a developer's perspective, the impact of cold start can be controlled through the configurable resources and careful choice of the programming language.

## 4.2 Instance Recycling Time and Lifetime

When a serverless function is first executed, the serverless platform creates the sandbox environment, loads the function's code in it, and executes the code. After the code has been executed, the sandbox environment is kept in a warm state for a certain amount of time (called *instance-recycling-time*) to serve any subsequent request for the same function. If

during that time, no subsequent request arrives, the sandbox environment is terminated so as to reuse the resources. A serverless platform may decide to terminate the sandbox environment after it has been in use for a certain period regardless of the usage. This time is called *instance-lifetime*.

Both *instance recycling time* and *instance lifetime* are very critical values to configure for not only the serverless platform but also for the users. A low value for these variables would mean that a serverless platform can free the resources quickly and re-purpose them for other applications while increasing the utilization of underlying resources, but for users, it can be devastating as the serverless functions would experience unnecessary cold starts hence degrading the performance of their serverless application. For a commercial serverless platform, it can lead to potential revenue loss by losing customers. While from the user's perspective, longer values would be ideal as their application would always find their serverless functions warm, hence reducing the latencies, but this may end up reducing the utilization of the underlying resource for the serverless platform [6].

For open-source serverless platforms [26, 111], a user can configure these values on their own and there have been studies suggesting using popularity analysis to configure these values on a per-application basis [73, 111]. But in commercial serverless platforms, these values are decided by the platform and a user has no control over the *instance-recycling-time* and *instance-lifetime*. There have been several peer-reviewed studies that looked at this aspect of commercial serverless platforms. Most of these studies followed a similar technique to infer the values for *instance-recycling-time* and *instance-lifetime*. Commercial serverless platforms allow a serverless function to use a limited amount of persistent storage for the time a sandbox environment is in use. Previous studies [87, 120, 122] use this storage to store an identifier for the serverless function when the function is invoked for the first time. Later they invoke the same function again and check if the identifier is still present; if it is not, then the sandbox environment was destroyed and the latter execution was done in a new environment. They show that different serverless platforms have different *instance-recycling* times, with Google Cloud Function having the longest of all (more than 120 minutes). AWS Lambda's recycling time is reported to be around 26 minutes. The authors could not find a consistent value for Azure Functions. Another recent study [17] claims this value to be 20-30 min for Azure Function, 5-7 min for AWS Lambda, and 15 min for Google Cloud Function. Hence, if a serverless function stays inactive for this *instance-recycling-time*, the subsequent request would incur an extra delay equal to a cold start.

In an independent study [37], the authors established a relation between *instance-recycling-time* and resources (i.e. memory) configured for the serverless function on AWS Lambda.

---

[6]Remember a user does not pay for idle times in serverless computing, hence this is a lose-lose situation for the serverless platform or cloud provider.

They found that a large value of memory configured for the serverless function tends to give it a small *instance-recycling-time*.

Regarding *instance lifetime*, in [122], using a similar technique, the authors found that Azure Function has the longest *instance-lifetime* as compared to AWS Lambda and Google Cloud Function. They also found that in the case of Google Cloud Function, the lifetime of an instance can be affected by the resources configured for the function. It is reported that *instance-lifetime* of an instance with 128 MB and 2,048 MB memory is 3–31 minutes and 19–580 minutes, respectively.

*Summary:* For a serverless function, *instance-recycling-time* is decided by the serverless platform. A serverless platform can employ more pro-active approaches to configure *instance-recycling-time* based on the application's popularity, as suggested in [111]. For an application developer, a low value for *instance-recycling-time* would affect performance by incurring extra cold-start delays. A developer can reduce the effect of cold starts by carefully choosing the language of the application and configurable resources.

## 4.3 Cost and Performance

The cost of cloud usage for *one execution* of a serverless function on a commercial serverless platform *p* can be calculated as follows:

$$COST_{per\_exec} = T(m) \times C(p,m) + G(p) \qquad (1)$$

where $T(m)$ is the run time of the serverless function given resources $m$ and $C(p,m)$ is cost per unit time of resources $m$ from the platform $p$. $G(p)$ denotes the fixed cost such as API gateway for AWS Lambda; if there is no fixed cost, $G(p)$ can be considered zero. Equation (1) shows that the cost of cloud usage directly depends on the run time of the serverless function and the price per unit time for resources $m$ [7, 11, 21, 25]. Hence all the factors that can impact the run time of a function will also impact the cost of cloud usage. To observe

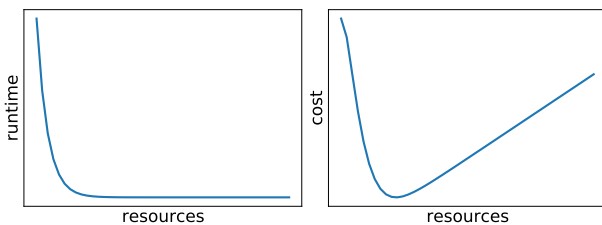

Figure 2: Performance and cost on AWS Lambda

the effect of configurable resources on the performance of a serverless function, we deployed various (I/O-intensive, memory-intensive, and CPU-intensive) functions on AWS Lambda and invoked them with varying resource configurations *e.g.* memory (more details in [44]). We show the

observed trends in the performance and cost with respect to the resources in Figure 2, across all function types. It can be seen that more resources lead to faster execution of the serverless function but the performance gain is limited after a certain point. Note that the performance of I/O-intensive and CPU-intensive functions also improves with more memory allocation. It is because of the fact that AWS Lambda allocated CPU share in proportion to the memory allocated [4]. This observation also confirms previous findings made in [44, 64, 93], which report a similar effect of resources on the performance.

Other factors that can affect the performance are summarized below:

**Cold Starts:** A serverless platform may decide to terminate the sandbox environment if it has been inactive for a certain amount of time, as explained in Section 4.2. Hence, serverless functions with less frequent invocations may incur the extra latency of cold start.

**Concurrency:** Previous studies [85, 87, 88, 122] looked at the effect of concurrency on the performance of serverless functions and found that the performance can be negatively impacted by a higher concurrency level. This is due to the particular resource provisioning policies of the serverless platforms as reported in [122]. In particular, AWS Lambda and Azure Function try to co-locate the function instances, hence causing more contention for resources. Recent work [108] shows that concurrency configurations can also impact the performance of serverless functions running on the open-source serverless platform Knative [27].

**Co-location:** Previous studies [44, 122] show that co-location of serverless functions on the same underlying resource can also result in significant performance degradation. Our preliminary experiment on OpenWhisk also confirms these findings. This is due to the fact that multiple function instances hosted on the same VM compete for resources such as disk I/O and network bandwidth, and this competition can adversely affect performance.

**Underlying Infrastructure and Policies:** As discussed in Section 4.6, the underlying infrastructure of commercial serverless platforms consists of diverse resources, and in addition to that, resource provisioning policies for the execution of a serverless function can also vary significantly from one platform to the other [122]. Hence, these aspects can also introduce significant uncertainty in performance.

Keeping in mind the tightly coupled nature of performance and cost of serverless functions, it is really important to find the "best" configuration of parameters (*online decisions*), *e.g.* memory, CPU, concurrency, such that they not only meet performance expectations but also optimize the cost of cloud usage. Previous approaches [44, 64, 68, 108] use various machine learning and statistical learning approaches to configure parameters, *e.g.* memory, CPU, concurrency and location, for serverless applications deployed on commercial and open-source serverless platforms. *We discuss these approaches in more detail in Section 7.3.*

*Summary:* The performance of a serverless function can be impacted by its configurable resources, choice of programming language, and the choice of serverless platform. The usage cost is calculated based on the configurable resources, the execution time, and the unit-time cost specified by the serverless platform.

## 4.4   Concurrency or Elasticity

Concurrency is the number of function instances serving requests for the serverless function at a given time. On-demand scaling by the serverless platforms – i.e. in case the demand for the serverless application increases, the serverless platform initializes more function instances to serve these requests concurrently – is one of the distinct features of the serverless computing model. Unlike IaaS, a user does not have to specify the scaling policies, rather the serverless platforms provision more function instances of the serverless function to cater to increasing demand. Most serverless platforms can scale up to a certain limit and en-queue any subsequent requests until one of the previous requests finishes execution and resources are freed. A platform's ability to scale quickly, and the maximum concurrency level that it can achieve, can be very critical to applications with fluctuating demand. To observe the maximum concurrency level that a commercial platform can support, Wang et al. [122] performed a comprehensive measurement study on three major cloud providers: AWS Lambda, GCF, and Azure Function. They found that out of all three, AWS Lambda was the best, achieving a maximum concurrency level of 200[7], while GCF and Azure Functions were unable to achieve advertised concurrency levels. FaaSdom [93], a recent benchmarking suite for serverless platforms, also found that AWS Lambda achieves the best latency in the face of an increased request rate for a serverless application – demonstrating its ability to quickly scale out. They also found that *one time decisions*, such as language and underlying operating system, can also affect the scalability of a serverless application. Another study [85] found that AWS Lambda and GCF perform better for varying demand when compared to IBM Cloud Function and Azure Function. We believe a platform's inability to scale well can come from the fact that scale-out is decided based on measured CPU load, a queue length, or the age of a queued message, which can take time to be logged. On the other hand, AWS Lambda launches a new function instance for a new request if current function instances are busy processing requests, as reported in [12, 85]. Using this proactive approach, AWS Lambda can scale out quickly without relying on any other measured indicator. As elasticity is one of the most advertised features of serverless computing, commercial serverless platforms are striving to improve their service

---

[7]This study was conducted in 2018. We believe higher concurrency levels can be achieved now given system upgrades.

by offering higher concurrency limits. AWS Lambda's recent documentation indicates that concurrency limits have increased significantly (>3000) and a user can request further increase [31].

Serverless platforms, such as Apache Openwhisk and Knative from Kubernetes, allow a user to configure a container-level concurrency limit, i.e. number of requests that a function instance can serve in parallel (where each request runs as a separate thread) [28, 35]. On the other hand, Azure Function allows a user to configure a maximum number of function instances that can be launched on a single VM to avoid the possibility of running out of underlying VM resources [13]. Schuler et al. [96] show that the container-level concurrency limit can affect the application's performance. They also suggest an AI-based (reinforcement learning) technique to configure the concurrency limit for Knative. The fact that a user can configure this particular concurrency limit on the fly also makes this limit an *online decision*. A user should be careful with configuring the container-level concurrency limit, as function instances running prior to making the configuration change will keep running with the old configuration (until terminated by the platform based on its settings), and only the new instances will assume the new concurrency limit. A user should wait for the system to be stable with the new configuration (i.e., all function instances with the old configuration are terminated) before making any further changes.

*Summary:* Serverless applications can elastically scale without any additional configurations. The maximum number of function instances that can run in parallel is determined by the serverless platform and can vary based on the cloud provider. Studies have found that among commercial serverless platforms, AWS Lambda scales best in terms of throughput.

## 4.5   CPU, Network and I/O

While using FaaS, a user can only configure certain parameters, *e.g.* memory, CPU-power, location, and concurrency, other resources such as CPU-, network- and I/O-share are decided by the serverless platform. In [122], the authors find that in case there is no contention, empirical results show that AWS Lambda puts an upper bound on the CPU share for a function with memory $m$ of $2m/3328$, while in the case of co-location, function instances share the CPU fairly and each instance's share becomes slightly less than, but still close to the upper bound. Similarly, Google also allocates the CPU share according to the memory allocated to the function. CPU allocation in proportion to memory assigned to a function is also specified in AWS Lambda and GCF's documentation [4]. Contrary to GCF and AWS Lambda, IBM Function does not allocate the CPU share in proportion to memory allocated to the function, as reported in [93], rather it keeps it constant as an increase in memory does not affect the performance of the function.

On the other hand, with Azure Function, the CPU share allocated to a function was found to be variable with the serverless function getting the highest CPU share when placed on 4-vCPU VMs. Note placement of function instances on VMs can be random from a user's perspective. In the case of co-location, the CPU share of co-located instances can drop. Similar to CPU share, disk I/O and network performance can also be affected by the resources configured for the serverless function and co-location, as reported in [44, 85, 122]. The performance usually improves when function instances are allocated more resources [123]. Measuring the network performance of FaaS platforms can be a challenging task considering the constantly changing network conditions and the multiple geographical regions that a developer can choose from, to deploy her serverless applications. A developer should opt for a geographical region that is closer to the intended users to reduce the access latency [123]. Our preliminary experiments also confirm this for the I/O performance, where the performance of I/O-intensive serverless functions improves when allocated more memory, as illustrated in Figure 2.

*Summary:* The CPU, Network, and I/O bandwidth of a serverless function can be impacted by the co-location of multiple functions on the same underlying resource (VM) and the instance placement policies of the serverless platform. An application developer can run various benchmarks (or consult measurement studies) to find the most suitable provider for her application.

## 4.6   Underlying Infrastructure

In a serverless computing model, a user only focuses on writing the code, and it is the serverless platform's responsibility to execute this code on any infrastructure/hardware. A user has no control over the underlying resources (types of VM where the application code would be executed). A developer may be interested in knowing the underlying infrastructure where their serverless application would be running to optimize the performance of their applications or to make other assumptions about the running environment of their application.

There have been several studies that tried to demystify the underlying virtual infrastructure for commercial serverless platforms. Lloyd et al. [87] discovered that serverless functions have access to the "/proc" file system of underlying VMs running the Linux operating system. By inspecting "/proc/cpuinfo", the authors discovered that the underlying VMs run Amazon Linux [1] and use CPUs that are similar to those of EC2 instances. Wang et al. [122] went one step further and using a similar approach, the authors conducted a wide study on all the big commercial serverless platforms, i.e. AWS Lambda, Google Cloud Function, and Azure Functions. They found that Google Cloud Function

successfully hides the underlying resources and the only information they could obtain was that there are four unique types of underlying resources. By inspecting "/proc/cpuinfo" and "/proc/meminfo", they found that AWS Lambda uses five different types of VMs having different vCPUs and memory configurations, mostly 2 vCPUs and 3.75GB physical RAM, which is the same as c4.large instances from EC2. The authors also noticed that Azure Function has the most diverse underlying infrastructure. While inspecting the contents of "/proc/*", they came across VMs with 1, 2, or 4 vCPUs, and the vCPU is either Intel or AMD model.

Knowing the underlying infrastructure can be helpful for developers to identify various performance-related issues. One example of that could be, a serverless function, running on Azure Function, placed on a VM with 4 vCPUs, can have more CPU share as compared to when placed on other types of VMs. Also, knowing the diversity of the underlying infrastructure can help the researcher explain the variability in performance for a given serverless platform.

*Summary:* Serverless platforms have diverse underlying infrastructure and this can introduce significant variability in the performance of a serverless function even when executed with the same configurable resources. Careful selection of the serverless platform by the application developer, and the usage of more pro-active approaches such as COSE [44] to dynamically configure resources for serverless functions, can mitigate this variability in performance.

## 5  Serverless Economic Model

As discussed earlier, FaaS platforms provide two main features: 1) the ability to scale quickly in the order of milliseconds without any additional configuration, and 2) a unique "pay as you go" pricing model, *i.e.*, a user only pays for the time the code is executing and not the idle time. The first feature helps with catering to bursty demands, *i.e.* when the demand suddenly increases for a short duration of time. In this section, we will have a detailed look at the latter feature and compare it with the traditional IaaS pricing model. IaaS services like Amazon's EC2 and Google's VM, have pricing models that not only charge based on minutes and seconds of usage but also have a different price per unit time as compared to their FaaS counterparts. In addition to the price factor, these VMs take extra labor to configure and maintain.

Previous works [44, 64, 68, 109] use various statistical, machine learning, and measurement-based methods to build the per-execution performance and cost model of serverless applications for a given platform. In this paper, we explain a more general analytical cost model, which in addition to per-request cost, considers the overall cost of deployment taking into account the demand.

Given the execution model of a serverless application for a certain serverless platform, pricing model, and demand (request per second), one can estimate the cost of deploying a serverless application on a commercial FaaS platform. Similarly, a user can calculate the cost of deploying a cloud application by renting VMs from a commercial cloud provider. In [16, 103], the authors present an economic model of deploying an application on commercial serverless platforms (FaaS), such as AWS Lambda, and compare it with the economic model when only IaaS resources (VMs) are used to deploy the application.

Specifically, the cost of FaaS based deployment can be described as:

$$COST_{FaaS} = \sum_{i=1}^{N}(COST_{per\_exec}) \qquad (2)$$

where $COST_{FaaS}$ is the total cost (per second) of running an application on a serverless platform. This cost depends on the rate of function invocations ($N$) and cost per execution from Equation (1), for platform $p$ and resources $m$ allocated to each request.

Similarly, the cost for IaaS based deployment ($COST_{IaaS}$) can be calculated as follows:

$$COST_{IaaS} = \lceil \frac{N}{VM_{r\_max}} \rceil \times C_{VM}(p) \qquad (3)$$

where $N$ is the request arrival rate, $VM_{r\_max}$ is the maximum number of requests that a VM can accommodate without violating SLO, and $C_{VM}(p)$ is the cost of renting a particular virtual machine $VM$ from platform $p$.

Note, in the above cost models, we do not consider the free tiers provided by the cloud provider. For example, AWS Lambda provides 1 million requests per month and 400,000 GB-seconds of compute time per month for free [7].

The key takeaways from the studies in [16, 103], following the cost models given by (2) and (3), are:

- FaaS platforms are cost-effective to deploy an application when the demand (request arrival rate) is below a certain threshold, referred to as Break-Even Point (BEP). Beyond BEP, IaaS resources are cheaper to use for their relatively lower cost per unit time.

- The authors also consider the different execution times and resources allocated to each request for the application on both IaaS and FaaS, and show that resources allocated for the execution of each request can also affect the value of BEP. Previous studies such as [44, 46, 64, 68] address the issue of finding the optimal resources for an application in the FaaS and the IaaS model.

The cost effectiveness of FaaS for low-rate and bursty computations has been observed and reported by various studies

[42, 66, 67] and has been leveraged by frameworks like LI-BRA [103] and Spock [76]. This unique FaaS economic model has revenue and performance implications for both the cloud provider and the application developer.

From the cloud provider perspective, serverless applications are stateless, run for shorter times and a user only pays for the actual utilization of resources. This is in contrast to traditional cloud applications which are generally stateful, run for longer times and a user pays for the lease duration irrespective of the usage. To maximize its revenue, a cloud provider may host multiple serverless applications from users on the same infrastructure and periodically evict applications based on its policies, as discussed in Section 4.2. Moreover, the particular pricing model allows a cloud provider to rent out infrastructure for as little as 1ms. FaaS offering also provides greater flexibility in terms of selecting/managing the underlying infrastructure, *e.g.*, operating system, communication protocols within the datacenter, security measures, and hardware types [67].

From an application developer's perspective, there can be multiple implications of using FaaS. First, there is an extra cost of building an application for FaaS as a traditional application may not work in this computing model. Moreover, depending on the provider, the development of serverless applications can differ significantly, *i.e.* code/dependency packaging is based on the runtime offered. So, a developer can suffer from vendor lock-in. Second, not only FaaS can be expensive if the demand stays high for a longer period of time, some platforms round up the execution time to, say, the nearest 100ms for cost calculation and a developer may end up paying for unused compute time, for example, if the function only runs for 50ms. Also, because of the provider's eviction policies, an application may experience frequent cold-starts, which in turn affect performance and cost. Lastly, FaaS platforms provide no strict SLO and an application can suffer from variable performance and outages [42, 107]. So, a developer should carefully consider all the performance and cost implications before opting for FaaS.

*Summary:* Serverless is more economical/efficient for applications with a low invocation rate and bursty demand. A developer should carefully anticipate the demand for her application and project the cost to decide whether FaaS is a cost-effective option for her application.

## 6    FaaS Usage

Even though serverless computing is a relatively new paradigm and still evolving, it has become a popular choice to deploy cloud applications. We believe that the following distinct features of serverless computing are the main reasons for its adoption and increasing popularity.

**F1** ***Development Model:*** FaaS allows developers to build cloud applications in high-level languages and provides API/CLIs [29, 38] to package code, along with dependencies, and to deploy it on the platform, thereby facilitating CI/CD (continuous integration and continuous delivery). Most platforms also provide integrated logging systems [5, 9, 19, 24] for debugging and monitoring.

**F2** ***No Back-end Maintenance:*** The serverless computing model offloads all back-end management from the application developer to the FaaS platform, which is responsible for the set-up and maintenance of underlying resources as well as scalability.

**F3** ***Pricing Model:*** As mentioned earlier, FaaS platforms offer a unique "pay as you go" pricing model. A user does not pay for deploying their application or for idle times. On the other hand, in an IaaS model, if a user has rented a VM, she pays regardless of the usage.

**F4** ***On-Demand Scalability:*** Unlike IaaS, where a developer has to configure scaling policies, serverless platforms assume the responsibility of scaling an application in case there is an increase in demand.

**F5** ***Quick Provisioning:*** Serverless platforms use advanced virtualization techniques, such as containers, to provision new instances of the application, which can be provisioned in the order of 10s of milliseconds [45, 54, 63, 100, 113, 122]. This feature allows a serverless application to scale out, in case of increasing demand, without suffering from performance degradation.

### 6.1    Interesting Use Cases

Considering the above development, cost, performance, and management advantages, FaaS is becoming a popular service to deploy cloud applications. Developers have employed FaaS to deploy rather simple DevOps to full production scale applications [20, 65, 82, 110]. We only present here some of the interesting use cases of serverless computing/FaaS.

Malawski et al. [94] show that AWS Lambda and GCF can be used to run scientific workflows. Serverless computing can also be employed to solve various mathematical and optimization problems [50, 112, 124]. Moreover, on-demand computation and scalability provided by serverless computing can be leveraged by biomedical applications [80, 83, 84]. MArk [128], Spock [76], Cirrus [55] and others [81, 117] explore deploying various machine learning applications using FaaS platforms. The authors in [70, 121] leverage the higher level of parallelism offered by serverless platforms to train machine learning models. FaaS for its on-demand, cost-effective computation power and elasticity has also been explored to deploy stream processing applications [39, 89]. Video processing is one such example, where a user may want to extract useful information from an incoming video

stream (video frames), where for each new incoming frame a serverless function can be spawned. Sprocket, ExCamera and others [48, 71, 129] describe the implementation of video processing frameworks using serverless functions. Authors in [59, 101, 102] explore the possibility of using serverless computing for IoT applications and services. Yan et al. [125] use serverless computing to build chatbots. Aditya et al. [41] present a set of general requirements that a cloud computing service must satisfy to effectively host SDN- and NFV-based services. Chaudhry et al. [58] present an approach to improve QoS on the edge by employing virtual network functions using serverless computing.

*Next, we explain the development, deployment, and management challenges of a FaaS application using ML inference modeling as an example application.*

## 6.2 ML Inference Models using FaaS

Since the advent of serverless computing, there have been several efforts exploring the possibility of using this computing model to deploy machine learning applications.

ML inference models are one such application, where an application developer can deploy a pre-trained ML model. When a user submits her query through an API, the inference model runs in a cloud service and the result is returned to the user. For QoS purposes, the inference model should return the results within a certain amount of time hence the model execution has an SLO [75]. Traditionally, developers have employed IaaS and other specific services, *e.g.*, Azure MLaaS [34], and AWS Sagemaker [8], to deploy such models but recently, approaches such as MArk [128], Spock [76] and others [81, 117] show that FaaS can also be leveraged for such applications for its quick provisioning time and pricing model (pay per request).

To deploy such an application, a developer implements the model in a high-level language such as Python and submits this code to the FaaS platform along with any dependencies/libraries [F1]. Usually, these models are Neural Networks (NN) and pre-trained models are either packaged with the code or placed in the external storage (S3 in case of AWS). When a user submits the query, the code along with the dependencies is loaded into a sandbox, and in case of cold start, it only takes few milliseconds [F5]. A developer does not have to worry about the underlying resources where the sandbox is placed as its managed by the FaaS platform [F2]. Queries can be submitted via HTTP request or other methods such as storage event, CLI, SDK, and triggers allowed by the platform. This code then downloads the model from the storage, processes the query in a serverless function, and the results are returned to the user or trigger event. A developer does not pay for the idle time in case the demand goes to zero and precisely pays for the time the model is serving queries [F3]. Moreover, when demand increases, the FaaS platform increases the number of sandbox instances, and scales back

when demand decreases [F4].

During the development cycle, a developer has to make decisions related to the language as it can affect performance (cold-start and elasticity as mentioned earlier). A developer can also optimize the code by removing the extra dependencies and library code that is not being used by the application to improve the cold-start latency. Also, the queries have performance requirements (SLO), the amount of resources, such as memory and CPU, configured for the sandbox environment is crucial to get the desired performance. Lastly, based on the query arrival rate, after a certain point, FaaS may not be the most economical option to process these queries, hence using an alternate IaaS based deployment can save substantial costs for a developer. *We discuss these challenges and their possible solutions in the next section.*

> *Summary:* The main driving factors for serverless adoption are simple development/deployment model, quick-provisioning time, on-demand scaling, abstraction of back-end management, and true "pay as you go" pricing model. While serverless adoption is increasing, there are certain challenges that need to be addressed.

## 7 Developer's Challenges

In the previous section, we discussed the suitability of the serverless computing model for various classes of cloud applications. In this section, we will take a closer look at the challenges that a developer may face while importing their application into the FaaS model, and optimizations that can leverage the serverless computing model efficiently. We will particularly focus our discussion on the challenges that a developer can address with limited control (*one-time* and *online decisions*). We will also discuss the state of the art solutions suggested to tackle these challenges and what remains unsolved. The discussion and insights presented in this section can help both the developer and researcher to optimize the FaaS usage and build new tools.

### 7.1 FaaSification and Decomposition

FaaS development and computing models significantly differ from traditional IaaS models. Hence, to deploy a legacy application using FaaS, a developer has to translate the application into this unique model. Recently, there have been approaches such as [62, 104, 114] that aim to automate this process for applications written in various languages. As pointed out by Yao et al. [126], these approaches either work for selected parts of the application or fail to leverage some of the key performance benefits offered by FaaS. In particular, these approaches replace a selected part of an application with a Remote Procedure Call (RPC) and deploy the selected part as a serverless function. While helpful to quickly deploy legacy applications using FaaS, these approaches miss taking

advantage of the elasticity feature offered by FaaS. We believe that an ideal FaaSification tool should not only consider producing the FaaS counterpart of the application but also leverage the elasticity offered by FaaS platforms. For example, through static/dynamic code analysis, the tool should identify the parts of an application that can be parallelized and generate corresponding serverless functions.

*Now, given a FaaSified version of an application, how can a developer leverage multiple FaaS platforms to optimize performance and cost?*

FaaS platforms offer diverse features, *e.g.,* elasticity limits, supported languages, configurable parameters, pricing models, etc. Moreover, as we have seen in Section 4, these platforms have varying underlying infrastructure and resource provisioning policies [122]. As a result, the performance and cost of the same application can vary significantly across different serverless platforms. In [51], the authors show that serverless functions with different bottlenecks, such as memory and computation, may have an ideal serverless platform on which they perform the best. This shows that serverless platforms are not *one-for-all*. Considering an application, which comprises multiple serverless functions with varying compute, memory, I/O bottlenecks, one platform may not suit all of the individual functions. We suggest investigating this idea further, where automated tools may help developers decompose their application into multiple serverless functions and then find the ideal serverless platform for each serverless function. This may require a sophisticated tool to perform code analysis [53] and measurement tools [93, 127] which can benchmark serverless platforms for different kinds of workload/computations.

Moreover, serverless platforms allow users to configure resources for each component of an application (if deployed as separate serverless functions), which may not be possible for a monolithic application deployed over a VM. In [127], the authors show that decomposing a monolithic application into multiple micro-services, instead of deploying the whole application as one unit, can lead to significant performance and cost gains. The authors also show an example application where decomposition leads to better performance and less cost. We also believe that decomposing an application would allow developers to cost-effectively fine-tune resources for various parts of the application.

To the best of our knowledge, we did not come across any previous work that suggests decomposing monolithic serverless applications across multiple providers to optimize the cost or performance. Costless [68] is the closest approach that suggests deploying a serverless application split across two platforms (edge and core) but it assumes that the application is already decomposed into multiple serverless functions.

## 7.2   Code Pruning

Another optimization that a developer can introduce is to optimize the code of the serverless application and remove any extra dependencies or unused library code. Usually, developers package a whole library along with the code even when the application is utilizing a small portion of it. This can adversely affect performance — as the application code is loaded into the sandbox environment at run time, studies [15, 78] have shown that the code package size can affect the cold start. Moreover, because the sandbox environment has limited resources, pruning extra code can also improve the performance of a serverless application. Nimbus [57] is one such framework that performs code optimization for serverless applications written in Java. We believe that this approach can be extended to other more popular languages used for serverless applications such as Python and JavaScript.

## 7.3   Parameter Tuning

On commercial serverless platforms, a user can only specify limited configurable parameters, such as memory, CPU, and location, for a serverless function. In Section 4, we discussed that measurement studies show that these configurable parameters can affect the cost of cloud-usage and the performance of serverless functions. As serverless platforms do not provide any guarantee (SLO) on the performance of serverless functions, configuring the parameters becomes even more crucial to get the desired performance of an application and optimize cost.

There have been a number of proposals suggesting various offline and online techniques to configure these parameters. Costless [68], given a workflow consisting of multiple functions, proposes a technique to efficiently distribute these functions across the edge- and core-cloud while reducing the cost of cloud usage and meeting the performance requirement. This approach relies on (one time) profiling of the performance of a serverless function in the workflow under possible memory configurations. It suggests suitable configurable parameters (memory) based on the profiling results, however, it fails to capture the dynamicity of the execution model. In [108], the authors show that the per-container concurrency limit in Knative can affect the throughput and latency of serverless functions. They suggest a reinforcement learning-based approach to find the optimal concurrency limit for a given deployment of the application. Even though this approach is adaptive, it only targets configuring the concurrency limit, but as discussed earlier, other parameters such as memory, CPU, and location can also impact performance. Moreover, we noticed that the authors did not address the feedback delay issue, which for Knative, in our experience, can be up to several minutes depending on the configuration. Sizeless [64] uses resource-consumption data from thousands of synthetic serverless functions to build a representative performance model. Then, using the performance model and performance logs of the target function, it suggests the best memory configuration. This approach may incur significant

cost overhead for running thousands of synthetic functions to get the required data to build the performance model. This approach also requires changes in the serverless application to collect the performance logs and only targets configuring memory for a function written in Node.js and deployed over AWS Lambda.

We believe that an ideal configuration finder should be a feedback control system, as illustrated in Figure 1, It should continually monitor the performance of serverless applications and configure these parameters on the fly if needed. There are a number of challenges for designing such systems: 1) serverless platforms have varying underlying infrastructure, resource provisioning policies, sandboxing techniques, and every time a serverless function is invoked, even with the same configurable parameters, performance can vary based on the co-location of functions and underlying resources. This makes it hard to predict the performance of the serverless function; 2) Our experiences with GCF and Kubernetes Knative, show that there can be a significant delay in the feedback loop, i.e. after the configuration is changed and until the new configuration takes effect (up to minutes as mentioned in Section 4.4). This excessive feedback delay can lead to performance instability as the state of the system might change during that time; 3) The impact of the changes in allocated resources on the performance of a serverless function can vary depending on the underlying serverless platform. In our experiments, we noticed that while an increase in allocated memory/CPU improves the performance of a serverless function on AWS Lambda and GCF, it did not significantly affect the performance on Apache OpenWhisk and IBM Function. Maissen et al. [93] make a similar observation about IBM Cloud Functions.

COSE [44] is an online statistical learning technique to configure various configurable parameters for delay-bounded chains of serverless functions or single functions. COSE not only achieves the desired performance for a serverless application but also reduces the cost of cloud usage. It can capture the dynamic changes in the execution model stemming from co-location and variable underlying infrastructure. COSE can be easily adapted for other parameters and platforms because it works as a stand-alone system that requires no changes to the serverless application. While COSE addresses most challenges of parameter configuration, it considers similar input sizes across multiple function invocations. COSE may not perform well if the input of the serverless application has large variations as the input size can affect the execution time.

## 7.4   Multi-Cloud Usage

Serverless functions are executed in light-weight sandbox environments, which can be launched in as few as 10s of milliseconds. So, in case an application experiences a sudden increase in demand, it can seamlessly scale out to cater to the increasing demand. This is a feature of serverless comput-

ing that has been leveraged by previous approaches, such as MArk [128], Spock [76], and FEAT [99], to hide the SLO violations for cloud applications deployed using traditional cloud services such as VMs. These approaches redirect a portion of the demand to the serverless counterpart of the application while scaling up traditional cloud resources which can take up to minutes to start up. These approaches may improve the performance of an application by reducing the number of SLO violations during scaling, at the expense of introducing a substantial development cost for a developer to build the serverless counterpart of the application. To reduce the development cost, a developer can employ an automated approach to build the serverless version of the application, similar to the approach suggested in [62, 114]. Another limitation of these approaches is that they suggest a one time configuration of resources for the serverless version of the application, which can lead to variations in the performance as explained in Section 7.3. As the goal of such approaches is to reduce the SLO violations, this variation in performance can adversely affect the application.

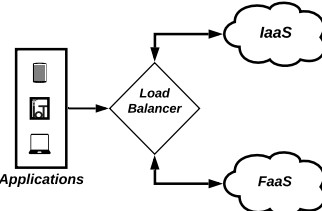

Figure 3: A balanced approach

We believe that in addition to performance, serverless computing also offers a unique pricing model, and as discussed in Section 5, serverless computing can be cost-effective for certain demand. For applications with large variations in demand, deploying them on VMs for the periods when demand is low can lead to sub-optimal cost. We propose to build a hybrid framework (Load Balancer, as illustrated in Figure 3) that leverages both aforementioned features of serverless computing, *i.e.* performance and economic model, by using serverless computing as: (1) an alternative to traditional cloud resources for a certain portion of the demand (consistently instead of only while scaling up VM resources), and (2) a fallback to the serverless counterpart of the application when demand is below BEP. To address the performance uncertainty in the serverless platform, we suggest that in addition to the multi-cloud framework, the developer should also employ more pro-active approaches, similar to COSE [44], to configure resources for the serverless counterpart of the application. COSE suggests the configuration for a serverless application that not only reduces the cost of cloud usage but also meets the specified SLO.

We also believe that serverless functions can be used as an alternative to VMs to offload the lightweight computations in a distributed application such as scientific workflows [94],

where small tasks requiring more concurrency and elasticity can be implemented as serverless functions while keeping the tasks with longer computation time and requiring more resources on VMs. One can leverage the "utilization" of the computation, i.e. how long the computation is and how often it needs to be executed, to decide whether the computation should be directed (and executed) over a dedicated VM or a serverless platform. The problem is how to optimally distribute computations to minimize the total cost. This is a challenging problem given the inherent performance-cost tradeoffs: VMs are cheaper for high-utilization (long-running and frequent) computations, on the other hand, serverless platforms are cheaper for low-utilization (short-running and less frequent) computations and have the advantage of elasticity.

Finally, developers have indeed started to leverage services from different cloud providers. A case study is presented in [20], where an invoicing application is built using various best-in-class services from different commercial cloud providers. The application is built using Google's AI and image recognition services along with two of Amazon's services (Lambda and API Gateway).

*Summary:* During the life cycle of a serverless application, a developer has to address various challenges starting from developing/porting the application in/to a relatively new programming model. To further optimize, a developer can also perform application decomposition and code pruning. She can also rely on various online/offline techniques to configure resources for her application to get the desired performance and optimize cost. Finally, depending on the usage, FaaS may not be the most economical option to run the cloud application, hence a multi-cloud scenario can help applications with fluctuating demand, without compromising on cost and performance.

## 8   Conclusion

Serverless computing has gained significant popularity in recent years. It offers an easy development model, back-end management, along with key performance benefits and a "pay as you go" pricing model. There is a significant amount of research articles addressing various aspects of serverless computing such as benchmarking/improving performance of commercial and open-source serverless platforms, new virtualization techniques for the execution environment, and studying the feasibility of serverless computing for a variety of cloud applications. In this paper, we look at these studies from an application developer's perspective and discuss how these studies can help her make informed decisions regarding her serverless application. We argue that serverless computing is becoming a popular choice to deploy various cloud applications for its distinct *cost and performance* benefits. While serverless adoption is pacing up, there are still a

number of challenges that need to be addressed. We identify potential challenges and open issues that must be addressed to make serverless computing a viable option to deploy cloud applications. We argue that pro-active approaches to configure resources for serverless functions can address the performance uncertainty issue, while frameworks to decompose serverless applications and to leverage various cloud services at the same time can reduce the operational cost as well as enhance the performance of cloud applications.

## Acknowledgement

This work has been supported by National Science Foundation Award CNS-1908677.

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
