# OpenReview forum: "SoK: Function-as-a-Service: From An Application Developer's Perspective"
_JSYS/2021/Mar_Papers — JSYS Mar 21_

### Official Review · AnonReviewer1 · 2021-03-15
**Review for "Serverless Computing: From An Application Developer’s Perspective"**

**Decision:**

Weak accept: good paper with flaws that can be fixed in three months

**Review:**

The paper presents a SOK on the subject of Serverless Computing, with the focus on the information that can be useful for application developers (as opposed to infrastructure managers). I must confess that I was unexcited about reading yet another Serverless SOK paper, but I found the perspective refreshing and enjoyed reading the (19-page!) paper.

The paper is structured in three big parts. In the first part, the authors describe the main results of several measurement studies with the goal of helping developers with deploying and configuring their serverless applications. In the second part, the authors discuss the performance vs. cost trade-offs and present a set of potential applications that can leverage the performance, cost, or both aspects of serverless computing. In the third and shortest part of the paper, the authors discuss future research directions for serverless computing.


PROS:
* The ideas are presented very clearly and the paper is well written and easy to read.

* The insights of the authors are well described and clearly identifiable. Furthermore, *most* are on-point for developers (i.e., they provide useful, concrete information that developers can use to improve their serverless applications, reduce their costs, or both).

* Tables 2 and 3 are novel and a very good contribution for a SOK paper.

* The list of papers cited by the authors is well curated (however, I do think they've missed a small number of important--as in very relevant--papers, as I describe later in my review).


CONS:

* At times, I felt like the authors got confused about what is actionable for developers and what isn't. It is good that developers have a SOK paper where they can read about research directions in serverless computing, but I think the authors should make a clear distinction between the things it is good that developers know about, and those that are actually things that developers can apply / use in practice.

* Similar to the issue above, some parts of the paper are aimed towards researchers and not application developers (for example, describing possible research areas). I feel this is fine as long as it is clear in the paper which parts are for the interest of application developers and which are aimed at researchers that wish to focus their ideas on things that would be of use to developers of serverless applications. For example, future research section 6.3 is something that--even though developers could tackle themselves--is very unlikely that they would. Serverless is not a common choice due to costs (not main reason) but rather due to ease of use; specifically, not having to worry about managing the infrastructure. Idea 6.3, though a very good idea, is not somethings that application developers would be interested in doing (it basically adds servers back to serverless!) but rather something that should be offered by providers and also perhaps explored by academia.

* Some important references are missing, see the next section of my review.

* I feel that the use cases section (Section 5) is not thorough enough; e.g., it is missing important use cases like: DevOps / Maintenance tasks, HTTP APIs that interact with end user, and ETL tasks. Furthermore, I think it distracts the reader from the most important parts of the paper (Sections 3, 4 and 6). Deciding to read a 19-page paper can be daunting; for this reason, I think removing Section 5 and instead pointing the reader to other studies about serverless applications (e.g., [1]) may be better. However, if you do think the section is critical and want to keep it, then you should at least add an "Other applications" subsection and briefly mention other important use cases that developers should be aware of.


Missing references:

* In Section 3.1, the authors talk about how serverless platforms use "approaches [that] employ container reuse, loose isolation between function instances, and memory snapshotting and restoring to achieve a cold-start latency that is as low as 10 ms or less." Here, the authors are missing approaches related to placement/routing/scheduling that are also used to achieve low cold-start latency (by increasing locality and thus maximizing environment re-use or benefiting from caching, in combination with placement/routing/scheduling algorithms). Examples of these are sticky routing [2], package-aware scheduling [3], and ENSURE [4]. There are also works related to reducing cold starts via optimal capacity planning like COCOA [5]. Not related to cold-starts but related to improving performance via improving locality, is the work of Bhardwaj et al. [6].

* In Section 3.5, you should also cite the AWS results by Tarasov et al [7] (see Figures 8 and 9 and their explanation in the paper), who also conducted similar measurement studies.

* Section 4 (Serverless Economic Model) as this is a SOK paper, I think you should cover more works that have been published on this topic. For example, the work by Eismann at al. [8] and the other papers they reference in their Related Work section (refs [2], [17], [39], [13], [18], [23] in their paper, though I think 1-2 of these you already have).


Minor nitpicks (not reasons to reject paper but would improve final manuscript quality if fixed):

* It is better if you don't use references as nouns. Example of one place where you are using references as noun: "Tools such as [48,49,61,98]
were built to help users...". You can change to "Tools like X, Y and Z [...]" or "Other have proposed tools that... [...]".

* Use dashes to join words when the words are used as adjectives, not when used as nouns. Examples that should be fixed: "one life-cycle of an application into two categories: one-time decisions and online-decisions"; in this example, life cycle and online decisions should not have dashes. Ref: https://apastyle.apa.org/learn/faqs/when-use-hyphen



References that the authors should consider citing:

[1] Serverless Applications: Why, When, and How? @ IEEE Software ( Volume: 38, Issue: 1, Jan.-Feb. 2021)

[2] Firecracker: Lightweight Virtualization for Serverless Applications @ NSDI 2020

[3] Beyond Load Balancing: Package-Aware Scheduling for Serverless Platforms @ CCGRID 2019

[4] ENSURE: Efficient Scheduling and Autonomous Resource Management in Serverless Environments @ ACSOS 2020

[5] COCOA: Cold Start Aware Capacity Planning for Function-as-a-Service Platforms @ MASCOTS 2020

[6] On the Impact of Isolation Costs on Locality-aware Cloud Scheduling @ HotCloud 2020

[7] Infinicache: Exploiting ephemeral serverless functions to build a cost-effective memory cache @ Usenix FAST 2020

[8] Predicting the Costs of Serverless Workflows @ ICPE 2020.


**Expertise:**

Actively publishing in this area

**Useful:**

yes

---

### Official Review · AnonReviewer2 · 2021-03-16
**Interesting content but needs work structuring it**

**Decision:**

Weak accept: good paper with flaws that can be fixed in three months

**Review:**

# Summary
This paper describes the state of the art of serverless from the perspective of a user/developer (as opposed to the more traditional provider perspective).
The collects the evaluation of aspects like the cost and the performance from other papers.
It also poses some open research challenges in the area and proposes high-level solutions.

Overall, the paper is useful and gathers the relevant state of the art.
My main concern with the paper is its structure which mixes high-level description, current research, and future work.
First, I give details on these general comments and then follow with comments for each section.

## Structure
The structure of the paper needs some work.
The main issue is the mix of current work and future work throughout the paper.
The measurements section also mixes actual content (measurement analysis) and description of the platform (belonging to the background).
In each section, I am leaving some proposals on how to improve this.

## FaaS
Given that the paper focuses on building applications on serverless, I think that FaaS should be brought up earlier.
Currently, FaaS is only instroduced when discussing the economical aspect (Section 4).
However, this should be clarified earlier on (maybe even in the introduction).
I would even argue that most of the topics discussed in the paper are more related with FaaS than with Serverless.

Everybody has a definition to serverless and FaaS which is fine as long as the definition is clear.
From this paper perspective, I would say that the serverless aspect refers to developers not having to care about managing servers and the programming mode lwhile FaaS refers to the pricing scheme.
Other variations (e.g., FaaS being the programming model) can be valid and up for interpretation but it should definetely be established.

## Function communication
The paper focuses on the execution of the functions but seems to overlook the communication aspect (across them and state with inputs and outputs).
It would be good to go over what are the options for communication (e.g., messages and distributed storage) and how users would choose which approach to use and why.


# Section 1: Introduction

## Contribution
The introduction, states what "an ideal SoK paper should address".
I do not think making this passive helps.
I would suggest to directly state what this paper will address and state it as its own contributions right away.
An alternative would be to talk about the previous and then go into what this paper does without passively bringing it up earlier.

## Minor comments
Note that SLA also refers to the penalties not only the guarantee (in contrast with SLOs).


# Section 2: Background
The end of the background section has forward pointers which transform the section into a motivation for Section 6.
It might be better to make this split more explicit.
In addition, it could use more structure (i.e., paragraph labels) to make the topics tackled easier to follow.


# Section 3: Measurements

## Structure
The beginning of Section 3 presenting the concepts of online and one-time decisions seem to belong to the background section.

In Section 3.2, when explaining performance, there are a few basic explanation that seem to belong to Section 2.
It is also questionable if Section 3.2 should be two separate sections for cost and performance.
The concepts are related but there is a clear separation; separating these two section would help following the ideas.

Section 3.2 talks about cost and then Section 4 goes into the economics.
It would be good to consolidate and structure in a way that makes the paper flow into just one description of the cost model.

Another example of the flow of this section not being clear is the amount of footnotes.
Some clarifications are useful and belong to a footnote.
However, most of the footnotes, should be introduced and incorporated to the main text.

Section 3.6 seems to also break the theme of the other subsection in Section 3.
I would expect it to be its own section (like the economic model) and probably earlier.

Overall, the content, flow and structure of Section 3 (specially cold starts) should be improved.
Most of the general concepts should be left in a shorter version of Section 3 and the rest (e.g., data related to Table 2) moved before Section 6.

## Cold starts
The concept of "cold start" is used in early section but it is not formally introduced until Section 3.1
The paper already cites [14] regarding the reclamation time.
However, [14] also includes thourough studies on cold starts which should be cited.

Section 3.5 is very related to cold start management and this link should be tighter explicitly referring to it and probably making sections 3.1 and 3.5 to follow each other.
This link is even mentioned in Section 3.2 when referring to cold starts.

## Table 1
I would expect a deeper introduction of Table 1 in the text instead of referring to it 3 times.
At least discussing what are the most significant differences across platforms.
It might be a good idea to have a section describing each of the providers and describing them deeper highlighting their singularities (these nuggets are mentioned throughout Section 3 but summarizing them per provider would be valuable).


# Section 4: Economic model

## Formulas
I cannot see the value on defining the FaaS and IaaS formulas as equations.
Currently, they follow the model f(x, y, z,...) which does not bring any insight other than saying what the inputs are.
To present the inputs just plain text or a table comparing the two seems a better choice.
Going towards the formula approach, one would give more specifics on the actual relation across

## Summary
The conclusions to Section 4 should be a little more nuanced and include a discussion of the provider perspective.
In particular, given the observations of the previous sections, a low rate application is very expensive to maintain for the provider.


# Section 5: Uses
## Programming model
Another main feature for using serverless is the programming model itself.
It is true that this is closely related to the ease of maintenance, scalability, etc.
However, the programming model is the enabled of most of this and there are users that do not care about these aspects but choose serverless and FaaS because of how easy it is to just code a function with clear inputs/outputs.
There are some references to this throughout the text but the introduction of Section 5 should include this for completeness.
This is even mentioned in Section 5.1 but this seems to be at the same level as the features mentioned in the introduction of Section 5.

## Other work
When describing the applications and uses, [64] does a pretty good job at it.
This section should at least refer to this work.

## Summary
The summary at the end of Section 5 is a little out of place.
It may need a conclusion section which compares or summarize the uses of serverless and then maps it back to the features.
Otherwise, the current summary just relates to the intro of Section 5 and the other subsections do not lead to anything that strengthens the summary.
At this point, it looks like the summary is just there to have the same structure across sections without a real purpose other than consistency.


# Section 6: Future research
## Structure
I would suggest to split the content of the paper into two clear parts: current state and the proposal.
Right now, the paper keeps pointing into the future work discussed in Section 6.
Section 6 can be presented in the introduction to describe the paper overall but there is no need to keep referring to Section 6 later on.
Currently, these references divert from the core which focuses on the state of the art.

In Section 6 itself is a little confusing as it is labeled as future research when it actually cites plenty of work already happening.
The parameter tuning section is a pretty good example of this.
I would differentiate clearly that (1) there is work already going to improve parameter tuning, (2) this work falls short in some aspects, and (3) there is some new techniques that should be explored (which this would actually be the future research).
This could be done by splitting each section into two or three (following the previous points) or having a section describing the current work on these areas and a separate section with a pure future research section.

**Expertise:**

Actively publishing in this area

**Useful:**

yes

---

### Official Review · AnonReviewer4 · 2021-04-09
**Good first part, improvement potential for second part**

**Decision:**

Weak accept: good paper with flaws that can be fixed in three months

**Review:**

Paper summary
-------------

This SoK paper in the area of serverless computing systematizes measurement studies, summarizes a serverless economic model, discusses challenges for different classes of applications, and outlines future research directions. It primarily takes the perspective of an application developer (or cloud user) focusing on decisions a developer can make to influence the performance and cost of serverless applications.

Strengths
---------
+ Strong focus on controllable parameters by developers linked to experimental studies (i.e., Table 2 establishing causal relationships). The refinement of this classification into one-time and online decisions is helpful.
+ Section 3 on Measurement Studies is well developed combining a useful classification with the discussion and comparison of detailed results from multiple studies. It covers the most important studies in the field.

Areas for improvement
---------------------
- The second part of the paper (Section 4-6) could benefit from clarified purpose, improved structure, and enhanced support from existing literature (see primary comments below).
- The flow of the paper could be improved by reducing redundancies and keeping the writing more focused/concise

Conclusion
----------

The paper addresses a relevant and active area of research where lots of papers have been published in the past few years. While there exist (multiple) SoK papers for the covered topics (measurement studies, serverless usage, future research), it presents the broadest view on serverless computing while also summarizing detailed technical results.

I like the paper and have some suggestions to strengthen its second part.

Primary comments
----------------

1. p3: The background information on serverless platforms (i.e., Table 1) is incomplete/outdated and inconsistent and should be verified carefully:
      1. Google Cloud Functions supports additional runtimes (Java, .NET, Ruby) as documented here: https://cloud.google.com/functions/docs/concepts/exec
      2. IBM lacks for example Ruby: https://cloud.ibm.com/docs/openwhisk?topic=openwhisk-runtimes
      3. Mixing languages with runtimes (Node.js vs C#). The .NET runtime is not limited to C# and hence misses other languages (e.g., F#). I recommend focusing consistently on runtimes instead of programming languages here to cover this aspect.
      4. The billing interval for AWS Lambda is outdated. Now 1ms (https://aws.amazon.com/lambda/pricing/). The argumentation on Page 4 and 11 about fixed cost per request might need to be adjusted or re-phrased accordingly.
      5. Please carefully verify all properties of Table 1.
      6. Optional: It would be great to provide traceability (using footnotes or an online appendix) because these properties are subject to frequent changes.
2. I like the paper's focus on a developer's perspective but have the question: Why do you think previous papers are written from the perspective of the service provider? I suggest clarifying or adjusting this argumentation.
      - Castro et al. [45] discuss the level of control a developer has (e.g., Table 1) and generally talk about "developers" often (e.g., about tools, frameworks, level of control)
      - Baldini et al. [40] similarly discuss developer control in serverless computing (see Fig. 2) in detail. I would argue that the developer's perspective is very much highlighted in discussions around the programming model as well as in many open questions they raise under "Open Research Problems". For example: "Can legacy code be made to run serverless?" is closely related to Section 6.2 in this paper on "Decomposing Serverless Applications".
      - Leitner et al. particularly focus on the developer's perspective in "A mixed-method empirical study of Function-as-a-Service software development in industrial practice"
3. p9ff: Section 5 on Serverless Usage needs some restructuring and/or clarification of its purpose:
    1.  Section 5 aims to "identify suitable classes of applications [...]" (p2) and "look at various classes of applications that are best suited for the serverless computing model" (p9). However, in my opinion, Section 5 discusses interesting usage scenarios of serverless (as the section title suggests) in research rather than identifying "best suited" or common application classes. The language used to describe these scenarios also indicates the more experimental nature of the cited studies rather than serving as examples for best suited applications:
        + potential: "the potential of using serverless computing for scientific workflows"
        + can: "serverless computing can be employed to solve various mathematical and optimization problems"
        + explore: "explore deploying various machine learning applications using serverless platforms" and "explored to deploy stream processing applications"
        + feasibility: "look at the feasibility of using serverless functions for IoT devices"
      One idea would be to adjust the formulation of the purpose for Section 5 more towards interesting usage scenarios that leverage serverless features. The discussion of challenges and open issues would also fit better rather than contradicting a "best fit" application class. The goal stated for an ideal SoK paper in the introduction also fits better than the current formulations in other parts of the paper: "2) what makes serverless com- puting ideal for certain classes of applications"
    2. Related to the previous comment, "5.5 Improving QoS of Cloud Applications" rather sounds like a general usage scenario than a class of application. Maybe a different subsection header (offloading, hybrid, idk) could alleviate this concern as I have the impression that "Improving QoS of Cloud Applications" doesn't fit well together with scientific workflows, ML and data processing and IoT.
    3. The connection between "distinct features of serverless computing" at the beginning of Section 5 with the remainder content on serverless usage should be clarified. The goal stated for an ideal SoK paper in the introduction gives a better hint towards this connection than provided in Section 5. A short note on how you derived these distinct features could be helpful given that other SoK papers [97,45] present similar key features. Could it be an option to move the description of "distinct features of serverless" to the background section instead?
    4. The Summary of Section 5 covers the wrong section and needs to be moved.
        1. The second part of the summary (i.e., "An application developer [...]") belongs to Section 6 and needs to be moved and adjusted to the ordering of Section 6 accordingly.
        2. The first part of the summary covers the authors' interpretation of distinct features of serverless computing. It would be preferable to have a summary that covers the key aspect of this section (i.e., serverless usage based on findings from literature).
4. Section 4 (p8f) on "Serverless Economic Model" would benefit from more diverse input and a unified model:
    1. In contrast to other sections of the paper, the economic model summarized here appears to be mainly a summary of a single blogpost (reference 13). I highly recommend enriching this discussion with additional perspectives from published literature. Some starting points could be:
        - Adzic and Chatley 2017: "Serverless Computing: Economic and Architectural Impact"
        - Eivy 2017: "Be Wary of the Economics of "Serverless" Cloud Computing"
    2. Why does the paper present two similar but different economic models for serverless computing (Section 3.2 Cost and Performance and Section 4)? Can these models be unified? If not, I suggest explaining the reasoning and motivation.
5. Section 6 (p12ff) on Future Research could benefit from a better distinction of challenges (what is the problem? maybe emphasized as question), existing work, and open issues (what remains unsolved?). I noticed its strong focus on discussing related work, which partially solve the challenges:
    1. 6.1 Parameter Tuning: COSE is praised to address (most of) the raised challenges, hence a clarification of what remains unsolved is recommended here.
    2. 6.3 Multi-Cloud Usage: The distinction between Section 5.5. and Section 6.3 needs to be much clearer. It appears counter-intuitive to describe such frameworks and approaches as ideal/common application class in Section 5.5. and then reference Section 6 for further discussion of them under Future Research as well. Reducing the redundancies between these sections could also help to keep the discussion more focused.


Detailed comments
-----------------

6. As a SoK paper covering multiple topics where other SoK papers exist, it could strengthen the diversity and credibility by considering existing views. For example:
    1. How do future research challenges compare to existing SoK papers? e.g., Eyk et al. [97] raise several issues concerning developer experience. Further reaching idea: Synthesize future work from primary research papers.
    2. Are there other classes of applications / usage scenarios that are relevant? What about web APIs? Clarifying the purpose might be sufficient to dismiss this point because I think the usage scenarios you discuss are more interesting (but not exhaustive).
7. I would argue the paper focuses primarily on the developer's perspective but not exclusively. The introduction could potentially hint towards this that some findings are also relevant from a service provider's perspective.
    1. For example: Section 3.1 says that "1) For serverless platforms: Serverless platforms can im- prove the cold-start latency by having fast sandboxing tech- niques or by keeping the sandbox instances warm for a longer time." before talking about the developer's perspective in point "2)".
    2. Given you also consider open-source platforms: (How) Does the developer's perspective differ compared to hosted platforms? For example, do you consider tunable system parameters of open-source platforms in control of developers (i.e., as a serverless user) or out of control (but still controllable by sys. admins)?
8. The introduction mentions to "augment [previous work] with our own experimental results and insights." In the remainder of the paper, own (preliminary) experimental results are mentioned about 4 times as a side note. Could it be helpful for the reader to mention how these "own experiments" are related to this paper?
9. I suggest introducing FaaS shortly and how it relates to serverless in the background section given you use both terms in the paper.
10. p13: "To the best of our knowledge, we did not come across any previous work that suggests decomposing monolithic serverless applications to optimize the cost or performance." => Related work published in IEEE Software: Ristov et al. 2021 "DAF: Dependency-Aware FaaSifier for Node.js Monolithic Applications"
11. Overall, reducing some redundancies could improve the flow and keep the paper more focused. For example:
    1. p13: Cost and performance benefits of serverless have been repeatedly discussed before so I suggest removing this sentence: "We believe that in addition to performance, serverless com- puting also offers a unique pricing model, and as discussed in Section 4, serverless computing can be cost-effective for certain demand."
    2. p12: The limitation of "limited control" has been introduced in the background and extensively discussed in Section 3. Hence, a re-explanation in 6.1 might not be necessary anymore: "In a serverless computing model, a user has limited control over the function’s run-time environment, i.e. hardware, op- erating system, CPU-type, etc."
12. p2: "The output of the serverless function is then returned as the response to the trigger." => This description is limited to synchronous function invocations and does not hold for asynchronous invocations. Either indicate that this sentence describes a typical case or explain different invocation types.
13. p2: "A user can control limited configurable parameters, namely memory, CPU-power, and location." => maybe a add clarifying modifier (e.g., such as memory size, ...) and/or mention that configuration capabilities differ per platform (e.g., memory is not explicitly configurable for Azure)
14. p9: "Summary: Serverless is more economical for applica- tions with low rate and bursty demand." => "bursty demand" is not explicitly discussed in Section 4. I would expect that a term used in the summary is at least mentioned in the previous section.
15. I suggest emphasizing that many of the specific results (e.g., CPU type, CPU share, instance lifetime, concurrency level) are of temporary nature and subject to changes (e.g., if providers decide to update hardware, policies, etc).

Minor comments (e.g., typos, readability, minor inaccuracies)
--------------

* p1: "introduced by Amazon in 2014 as Amazon Lambda [1]" => "AWS Lambda" (AWS != Amazon)
* p1: "Tools such as [48,49,61,98] were built" => preferably re-formulate more self-containing or de-emphasize the citations alike "Many tools [...] were built to ..."
* p1 "allows a developer to focus on writing code in a high-level language (as shown in Table 1)" => could clarify reference to Table 1 as listing examples of runtime languages (rather than showing how it allows a developer to focus on writing code)
* p1 "Hellerstein et al. and Jonas et al. [40, 60, 64] also provide" => Please correct the order and names of these citations
* p2 "Then we present an economic model of serverless computing and compare it with traditional Infrastructure-as-a-Service (IaaS), and iden- tify suitable classes of applications that can leverage server- less computing for its performance/cost (Sections 4 & 5)." => misses a word at the end of the sentence. I suggest clarifying that Section 4 covers the economic model by moving the section reference accordingly instead of grouping them.
* p3: The first paragraph of Section 3 is largely redundant with the 2nd last paragraph in Section 2 (just before). Consider reducing this redundancy (maybe shorten the background because it fits better into the flow in Section 3)
* p5, Table 2: Could it make sense to align the names in the first column with the subsection headers, maybe even link them? It could make it easier to relate the table rows to the individual subsections that currently have slightly different names.
* p5: "the underlying infrastructure of commercial serverless platforms consist" => the infrastructure (ie it) *consists*
* p7: "The authors could not find a consistent value for Azure Functions. While another recent study [14] claims this value to be 20-30 min for Azure Function, 5-7 min for AWS Lambda and 15 min for Google Cloud Function." => I would expect some contrasting argument for the 2nd sentence starting with "while". Maybe drop "While" and connect the sentences better.
* p9: "Serverless is more economical for applica- tions with low rate and bursty demand." => low invocation rates
* p9: "Even though serverless computing is a relatively new paradigm and still evolving, there have been several attempts from independent developers and researchers to deploy vari- ous applications using this computing model." => why only these groups? This sounds like serverless would be a niche offering but given the adoption by companies running production workloads, I suggest re-phrasing this sentence more optimistic.
* p10: "While ‘pay as you go‘ pricing, on-demand scaling, and minimal cold start, make serverless computing a good fit to deploy machine learning models" => no comma (,) after cold start
* p10: "Recent approaches [47, 83, 84]" => consider omitting recent here for a study [84] published 4 years ago (which is about half the lifetime of AWS Lambda)
* p13: Figure 4 could be presented horizontally more space-efficient and analogous to Figure 1


**Expertise:**

Actively publishing in this area

**Useful:**

yes

---

### Official Review · AnonReviewer3 · 2021-04-12
**SoK: Serverless Computing: From An Application Developer's Perspective**

**Decision:**

Weak accept: good paper with flaws that can be fixed in three months

**Review:**

Paper summary: This paper provides a systemization of knowledge (SoK) for serverless computing from a developers perspective. Besides providing an overview of how function-as-a-service (FaaS) platforms work, it characterizes existing  FaaS platforms along three dimensions: their features and configuration parameters that are exposed to a user via prior measurement studies, the applications across various domains that can and are suitable for serverless computing, and future research directions that can improve the utility of serverless computing from a developers perspective.

Strengths: The paper provides a comprehensive overview of research in the serverless computing domain, presented from a perspective that is useful for application developers. It is also well-written, with very helpful summary/conclusion boxes summarizing takeaways for each dimension of FaaS platform they analyze.

Areas of improvement: The discussion on characterization of prior studies on serverless computing provides detailed insights on what those studies found, but could benefit from a more detailed discussion on the why and how behind those findings.

Comments for authors: Thank you for submitting your work to JSys’21! I enjoyed reading the systemization of knowledge — it was a well written summary of prior studies on serverless computing, with a novel application developer-driven perspective. While the analysis of prior work is comprehensive, the summary boxes at the end of sections/subsections are helpful in highlighting the major takeaways from the many prior works in the space.

My main suggestion for improvement for this study is to focus a bit more on the why and how behind the findings of prior work presented in the paper — currently, the text primarily focus on only what the findings are. As a few concrete examples:
* In section 3.1, it is mentioned that the choice of programming language have performance implications — scripting languages have 100x less cold-start delays compared to compiled runtimes. What is the intuitive reason behind this, and how does this impact an application developer’s choice of language? Should they always prefer scripting languages to avoid cold-start delays?
* Again, in section 3.1, choice of  serverless provider is cited to impact cold-starts as well, depending on their underlying infrastructure, etc. Why such a difference, and how does it vary across different cloud providers? How should an application developer reason about this choice?
* In section 3.2/3.3, the summary of prior studies suggest that choice of underlying operating system, concurrency, co-location and underlying infrastructure/policies of cloud providers affect performance. Again, how do they affect performance, and why? How does it factor into an application developers choices?
* In section 3.4, there is a detailed description on how CPU and memory allocation is shared across various cloud providers, but the discussion on network I/O is brief — how and why exactly is network IO affected by resource configuration and co-location?
There are more instances through out sections 3-6, and could benefit from brief intuitive descriptions of the underlying reason behind presented findings, and a more detailed characterization of their impact on developer choices.

Second, I was a bit confused by Figure 2: is the plot obtained from real experiments, or simply a characterization? The figure is missing axes labels, and makes me believe it is the latter, although the text states “we deployed various (I/O-intensive, memory-intensive, and CPU-intensive) functions on Amazon Lambda and invoked them with varying resource conﬁgurations.” The figure is referenced for a number of resource types (CPU, memory, network, etc.) as well — do all the resources exhibit the same characteristic trend? How does it vary across different cloud providers?

Detailed comments:
* Your paper is titled ‘Serverless Computing: From An Application Developer’s Perspective’, but focuses only on FaaS platforms. There are other BaaS platforms that also claim to operate under a serverless model — if your focus is primarily FaaS, perhaps its better to reconsider the title.
* In Section 3.1, you mention “One has to be careful with conﬁguring more resources for the serverless function to remedy cold start, as it can increase the cost of running the serverless function.” I was expecting to see a characterization of this tradeoff there, but found it in Section 3.2 instead. Perhaps it might be useful to provide a forward pointer in 3.1 to 3.2.
* The discussion on reducing/circumventing cold-start latencies talks about how serverless platforms can reduce them. While this is informative, it isn’t particularly useful for an application developer, since they do not have any control over it. Given your focus on application-developer perspective, perhaps it can be dropped.
* Section 3.5 seems to focus on aspects very closely tied to cold-starts (Section 3.1). Is there a reason for separating the two? I feel merging the two sections would lend depth to cold-start analysis.
* I really liked the summary of applications in Section 5. One trend that was consistent in describing applications (which I really appreciated) was a discussion on why those applications are a good fit for serverless computing, and what challenges they face in existing platforms. I would suggest adding a table that summarizes these opportunities and challenges — it would really help summarize your discussion.
* IMO Section 5.2 is better titled as Data Analytics applications (which subsumes ML). Also, the section is missing a number of recent works on Serverless Analytics (e.g., Starling [SIGMOD’20], Cloudburst [VLDB’20], etc.). Might be useful to look at this years publications that appeared after your submission (e.g., Caerus [NSDI’21]).
* The paper mentions in number of places (abstract/intro, section 5, 6) that serverless platforms can lead to performance gains. Specifically, in Section 5: “In addition to the ease of development, the particular pricing model and on-demand elasticity of serverless computing can beneﬁt such applications both in terms of cost and performance.”  This sentence is contradicted two paragraphs later: “However, the stateless nature of serverless functions can adversely aﬀect the cost and performance of such applications.”   As I understood it, the performance gains stem from being able to scale and provision resources faster using serverless platforms, outperforming IaaS solutions when they are underprovisioned — however, it is unlikely that a serverless realization will outperform a sufficiently provisioned IaaS solution, at least today (e.g., due to cold-start overheads, shared infrastructure overheads, etc., as you mention in Section 3). It would be useful to disambiguate scalability+elasticity from performance for serverless platforms.
* Section 6 could benefit from similar conclusion/summary boxes employed in Sections 3-5

**Expertise:**

Actively publishing in this area

**Useful:**

yes

---

### Meta-Review · Area_Chair1 · 2021-04-15

**Recommendation:** Revise
**Confidence:** 5

**Metareview:**

I would like to thank the authors for submitting their paper to JSys. Your paper is highly relevant to our area (Serverless Computing), and I hope you would find the comments provided by the reviewers to be detailed, relevant, and constructive.

There is consensus among all four reviewers that your paper can be accepted after fixing a number of (mostly easily fixable) issues. Therefore I recommend a one-shot revision for your paper. **For your paper to get accepted, please consider ALL issues mentioned in the reviewer comments carefully and provide reasonable rebuttals for any comments not addressed.** Also, given that the next submission is going to be a few months after your initial submission, I kindly encourage you to include the latest relevant serverless papers in your survey paper.

The reviewer comments are very comprehensive, and I do not have major recommendations for you beyond their comments. However, on a minor note beyond what reviewers have already brought up, I noticed that in Table 1 you mention a 100ms billing interval for AWS Lambda. It was recently reduced to 1ms. Please update it to reflect the correct value. Also, please make sure you cite other relevant serverless survey papers. For instance, [1] is relevant but is not currently mentioned in the paper. The reviewers have mentioned other papers to cite that would enrich your survey paper.

JSys values your contribution and we enthusiastically look forward to your revised submission.

[1] "Function-as-a-Service performance evaluation: A multivocal literature review"

---

### Decision · Program_Chairs · 2021-04-15

**Decision:**

Accept

**Comment:**

The paper is much stronger after revision and highly appreciated by the reviewers.